# Ada2I: Enhancing Modality Balance for Multimodal Conversational Emotion Recognition

## ABSTRACT

Multimodal Emotion Recognition in Conversations (ERC) is a typical multimodal learning task in exploiting various data modalities concurrently. Prior studies on effective multimodal ERC encounter challenges in addressing modality imbalances and optimizing learning across modalities. Dealing with these problems, we present a novel framework named **Ada2I**, which consists of two inseparable modules namely **Ada**ptive Feature Weighting (AFW) and **Ada**ptive Modality Weighting (AMW) for *feature-level* and *modality-level* balancing respectively via leveraging both **I**nter- and **I**ntra-modal interactions. Additionally, we introduce a refined disparity ratio as part of our training optimization strategy, a simple yet effective measure to assess the overall discrepancy of the model's learning process when handling multiple modalities simultaneously. Experimental results validate the effectiveness of **Ada2I** with state-of-the-art performance compared against baselines on three benchmark datasets including IEMOCAP, MELD, and CMU-MOSEI, particularly in addressing modality imbalances.

## CCS CONCEPTS

• **Information systems** → **Sentiment analysis**; • **Computing methodologies** → **Discourse, dialogue and pragmatics**; • **Human-centered computing** → **Human computer interaction (HCI)**.

## KEYWORDS

Multimodal Emotion Recognition, Imbalance Modality, Adaptive Feature Weighting, Adaptive Modality Weighting, Disparity ratio

## 1 INTRODUCTION

Multimodal learning is an approach to building models that can process and integrate information from multiple heterogeneous data modalities [2, 20, 21], including image, text, audio, video, and table. Since numerous tasks in the real world involve multiple modalities, multimodal learning has become increasingly important and attracted widespread attention as an effective way to accomplish these tasks. In recent years, the field of Emotion Recognition in Conversations (ERC) has witnessed a surge in effective models [8, 26, 29]. Moving beyond unimodal recognition, the utilization of multimodal data offers a multidimensional perspective for more nuanced emotion discernment [9, 19, 24]. Consequently, the incorporation of multimodal data is a natural evolution for enhancing emotion

*ACM MM, 2024, Melbourne, Australia*
© 2024 Copyright held by the owner/author(s). Publication rights licensed to ACM.
ACM ISBN 978-x-xxxx-xxxx-x/YY/MM
https://doi.org/10.1145/nnnnnnn.nnnnnnn

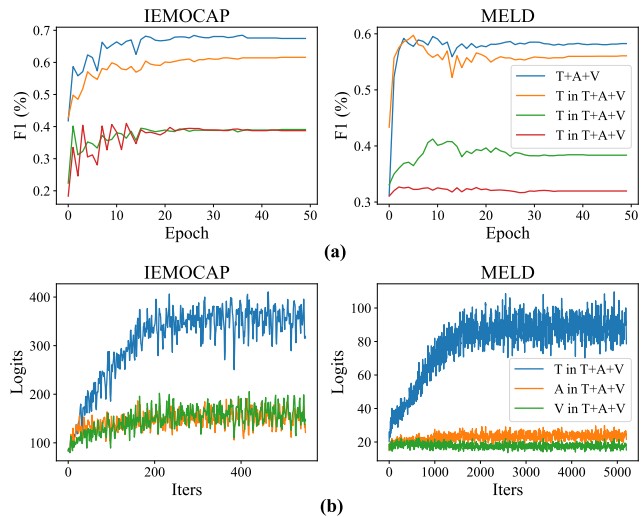

**Figure 1: (a) Weighted F1 scores for the multimodal setting (T+A+V) compared with each unimodal encoder, and (b) batch-average unimodal-logit scores.**

recognition in conversations. However, the widespread adoption of multimodal learning has revealed underlying challenges, with a primary focus on modality imbalances. These imbalances entail disparities in the contributions of individual modalities to the final decision-making process.

As illustrated in Figure. 1, the text modality quickly addresses the overall model performance and the joint logit scores, whereas the visual and audio modalities remain under-optimized throughout the training process. In addressing modality imbalance, diverse terminologies have emerged to characterize this phenomenon and explore its underlying causes. Terms such as "greedy nature" [38], "modality collapse" [15], and "modality imbalance" [6, 22] have been employed in various studies. These terms are associated with factors such as the "suppression of dominant modalities" [25], "different convergence rates" [35], "diminishing modal marginal utility" [36], or "modality competition" [14]. In essence, two primary perspectives emerge regarding this problem [36]: firstly, modalities exhibit varying levels of dominance, with models often overly reliant on a dominant modality with the highest convergence speed, thereby impeding the full utilization of other modalities with slower convergence speeds. Secondly, modal encoder optimization varies, necessitating the adoption of multiple strategies. Some approaches [7, 25] attempt to modulate the learning rates of different modalities based on the fusion modality. However, these approaches often overlook the impact of *intra-modal data enhancement* [44]. For instance, right from the initial representations through the modal encoder, the outputs can lead to misleading final results, resulting

in its weakened position across all modalities. Hence, from the outset, it is crucial to enhance representations for each modality, regardless of whether they are weak or strong, as it can affect the imbalance in learning across modalities.

Moreover, current methodologies primarily focus on interactions between pairs of modalities [6, 22, 25, 39], resulting in complex computations and inadequate treatment across all modalities. These methods are commonly applied in tasks such as audio-visual learning [25, 39] and multimodal affective computing [44], often using datasets related to sarcasm detection, sentiment analysis, or humor detection. However, there is a lack of methods explicitly tailored for multimodal emotion recognition in conversation, especially for well-known multimodal datasets like IEMOCAP [3], MELD [27], and CMU-MOSEI [1]. Additionally, in recent prominent studies [17, 32], while overall performance for multimodal emotion recognition tasks has notably increased, a closer examination of the "importance of modality" reveals that pairwise modalities consistently fail to achieve satisfactory performance, creating a significant gap compared to leveraging all three modalities simultaneously. Therefore, it is crucial to simultaneously leverage learning from all modalities while also significantly enhancing the capabilities of weaker modalities to improve the overall learning performance of multimodal emotion recognition models in practical applications.

In this paper, we propose a novel framework named **Ada2I** that addresses imbalances in learning across audio, text, and visual modalities for multimodal ERC. It consists of two primary modules including **Ada**ptive Feature Weighting (AFW) and **Ada**ptive Modality Weighting (AMW) for *feature-level* and *modality-level* balancing respectively in the consideration of **I**nter- and **I**ntra-modal interactions. Focusing on *feature-level balancing* using Adaptive Feature Weighting (AFW), we apply tensor contraction to infer feature-aware attention weights for each modality, which aims to produce a feature-level balanced representation for each conversation. As an important component of AFW, Attention Mapping Network controls the balancing via maximizing the alignment between unimodal features and their corresponding attention coefficients. For *modality-level balancing* using Adaptive Modality Weighting (AMW), we further exploit feature-level balanced representations from the preceding AFW module to generate modality-level balanced ones through modality-wise normalization of features and learning weights before being used to enhance the emotion recognition. Additionally, we utilize the concept of disparity ratio, although with modifications compared to the study by Peng et al. [25], called OGM-GE, as a value to supervise the training process and evaluate the model. Specifically, while OGM-GE [25] introduced gradient modulation for pairs of modalities, we refine it to handle all three modalities simultaneously—textual, visual, and audio—in the Multimodal emotion recognition in conversation task. This adjustment reduces model complexity and overall processing time, leading to enhanced efficiency.

To summarize, our contributions are as follows:

- We propose an end-to-end framework named **Ada2I** that addresses the issue of imbalance learning across modalities comprehensively for the multimodal ERC task. It not only considers modality-level imbalances but also leverages feature-level representations to contribute to the balancing step in the learning process.

- With two modules intricately designed yet inseparable, Adaptive Feature Weighting (AFW) is crafted to enhance the representation of each conversation at the feature level, while Adaptive Modality Weighting (AMW) is proposed to optimize the modality-level learning weights during training. Additionally, we redefine the disparity ratio, a simple yet effective measure, to assess the overall discrepancy of the model's learning process when simultaneously handling multiple modalities, rather than just two as in the original approach from [25].

- Our empirical experiments illustrate the effectiveness and enhancements provided by our method in comparison to existing approaches across three prevalent multimodal ERC datasets: IEMOCAP [3], MELD [27], and CMU-MOSEI [1].

The paper is structured as follows: Section 2 presents related work while we descibe the proposed framework Ada2I in Section 3. Experiment settings are explained in Section 4, and Section 5 covers the experimental evaluation and results. Finally, Section 6 summarizes findings.

## 2 RELATED WORK

### 2.1 Multimodal Emotion Recognition in Conversation

Multi-modal Emotion Recognition (ERC) has emerged as a focal point within the affective computing community, garnering significant attention in recent years. The integration of multimodal data provides a multidimensional perspective, enabling a more nuanced understanding of emotions. Moreover, researchers have increasingly turned to multimodal fusion techniques, combining text, audio, and visual cues to enhance multimodal ERC performance [9, 10, 16, 18, 24]. ICON [9] employs two Gated Recurrent Units (GRUs) to capture speaker information, supplemented by global GRUs to track changes in emotional states throughout conversations. Similarly, MMGCN [37] utilizes Graph Convolutional Networks (GCNs) to capture contextual information, effectively leveraging multimodal dependencies and speaker information. On the other hand, Multilogue-Net [30] introduces a solution utilizing a context-aware RNN and employing pairwise attention as a fusion mechanism. TBJE [4], adopts a transformer-based architecture with modular co-attention to jointly encode multiple modalities. Additionally, COGMEN [16] is a multimodal context-based graph neural network that integrates both local (speaker information) and global (contextual information) aspects of conversation. Moreover, CORECT [24] employs relational temporal Graph Neural Networks (GNNs) with cross-modality interaction support, effectively capturing conversation-level interactions and utterance-level temporal relations. GraphMFT [18] utilizes multiple enhanced graph attention networks to capture intra-modal contextual information and inter-modal complementary information. More recently, DF-ERC [17] emphasizes both feature disentanglement and fusion while taking into account both multimodalities and conversational contexts. Moreover, AdaIGN [32] employs the Gumbel Softmax trick to adaptively select nodes and edges, enhancing intra- and cross-modal interactions. While these methods primarily focus on designing model structures, they overlook the challenges posed by modality imbalance during multimodal learning.

## 2.2 Imbalanced multimodal learning

Despite the suggestion by [13] that integrating multiple modalities could enhance the accuracy of latent space estimations, thereby improving the efficacy of multimodal models, our investigation within the multimodal ERC task reveals a phenomenon contradicting this notion. The problem of modality imbalance persists as a significant challenge in multimodal learning frameworks, particularly in tasks like multimodal Emotion Recognition in Conversations (ERC). Conventional methods often prioritize one modality over others, assuming that certain types of sensory data are more relevant for a given task. For example, textual cues may receive greater emphasis, while visual or audio cues alone might be prioritized [16, 24, 37].

Current methodologies for addressing imbalanced multimodal learning primarily focus on tasks such as audio-visual learning with a focus on optimizing pairwise modality learning [6, 25, 39], sentiment analysis, and sarcasm detection [44]. However, these approaches often have task-specific limitations and framework restrictions, limiting their broader applicability. For instance, Wang et al. [35] identified that different modalities overfit and generalize at different rates, leading to suboptimal solutions when jointly trained using a unified optimization strategy. Peng et al. [25] proposed OGM-ME method where the better-performing modality dominates the gradient update, suppressing the learning process of other modalities. MMCosine [39] employs normalization techniques on features and weights to promote balanced and improved fine-grained learning across multiple modalities. Notably, there is a lack of specific approaches tailored for multimodal Emotion Recognition in Conversations (ERC) apart from the work by Wang et al. [36]. Recently, Wang et al. [36] observed a phenomenon referred to as "diminishing modal marginal utility" and proposed fine-grained adaptive gradient modulation, which was applied to ERC, while $I^2$MCL considers both data difficulty and modality balance for multimodal learning based on curriculum learning for affective computing, though not specifically for emotion recognition. To comprehensively address the challenge of modality imbalance in multimodal ERC, we propose an end-to-end model that ensures balance among text, audio, and visual modalities during training.

## 3 METHODOLOGY

In the context of a conversation $C$ with $N$ utterances $\{u_1, u_2, \ldots, u_N\}$, the task of Emotion Recognition in Conversations (ERC) is to predict the emotion label for each utterance in the conversation from a predefined emotion category set $\mathcal{E}$. Each utterance is associated with $M$ modalities, i.e. textual (t), audio (a), and visual (v) modalities, represented as:

$$u_i = \{u_i^t, u_i^a, u_i^v\}, i \in \{1, \ldots, N\} \tag{1}$$

where $u_i \in \mathbb{R}^{M \times d}$, $d$ signifies the dimension of modal features. For each modality $m$, we derive multimodal features $\{\mathbf{X}^m\}_{m \in \{t,a,v\}} \in \mathbb{R}^{d_m \times N}$ for the conversation $C$. Here, $\{d_m\}_{m \in \{t,a,v\}}$ is the feature dimension of each modality.

In this section, we outline our proposed model Ada2I, including its main sub-modules: Modality Encoder, Adaptive Feature Weighting and Adaptive Modality Weighting. We also refine the disparity ratio metric as part of our Training Optimization Strategy. The architecture of our model is shown in Figure 2.

## 3.1 Modality Encoder

Given a conversation $C$, a **Transformer** [33] network is utilized as the encoder to generate a unimodal representation $\mathbf{Z}^m \in \mathbb{R}^{N \times d_m}$ respecting to the modality $m$ as:

$$\mathbf{Z}^m = \phi(\theta^{(m)}, \mathbf{X}^m), m \in \{t, a, v\} \tag{2}$$

where the function $\phi(\theta^{(m)})$ is the Transformer network with learnable parameter $\theta^{(m)}$.

## 3.2 Adaptive Feature Weighting (AFW)

*3.2.1 Tensor-based Multimodal Interaction Representation.* Motivated by the tensor-ring decomposition method introduced by [42], we extend the traditional attention mechanism by replacing the query (**Q**) and key (**K**) representations with tensor-ring decomposition-based counterparts. This modification results in query tensor-ring representation $\mathcal{G}_Q$ and key tensor-ring representation $\mathcal{G}_K$, which facilitate the acquisition of more compact modality representations. Additionally, inspired by [31], we integrate a tensor-based multi-way interaction transformer architecture into our model. This enhancement allows the model to capture multi-way interactions among modalities, thereby enhancing its capability to discern intricate multimodal relationships.

We employ a tensor-ring-based generation function to retrieve the multi-interaction multimodal query tensor $Q$ and key tensor $\mathcal{K}$ from the input modality presentations $\mathbf{Z}^m$. Specifically, we compute $Q$ and $\mathcal{K}$ as follows:

$$\begin{cases} Q = \mathbf{Tr}\{\mathcal{G}_Q^{(t)}, \mathcal{G}_Q^{(a)}, \mathcal{G}_Q^{(v)}\} \in \mathbb{R}^{d_t \times d_a \times d_v} \\ \mathcal{K} = \mathbf{Tr}\{\mathcal{G}_K^{(t)}, \mathcal{G}_K^{(a)}, \mathcal{G}_K^{(v)}\} \in \mathbb{R}^{d_t \times d_a \times d_v} \end{cases} \tag{3}$$

Here, $\mathbf{Tr}\{.\}$ represents the tensor-ring decomposition function, which naturally provides the low-rank core tensor representations $\mathcal{G}_Q^m$ and $\mathcal{G}_K^m$ for each modality.

To perform multimodal attention in the tensor space, we need to compute the attention coefficient matrix, $\Theta$, from the tensorized input. To achive this, we can first compute the Tensor-ring Key representation and Tensor-ring Query representation of input data, $\mathcal{G}_Q^m \in \mathbb{R}^{d_m \times r_s \times r_w}$ and $\mathcal{G}_K^m \in \mathbb{R}^{d_m \times r_s \times r_w}$, where $m \in \{t, a, v\}$, the index $s, w \in \{1, 2, 3\}$, and $s \neq w$. The attention coefficient matrix $\Theta$ of modality $m$ is formulated as follows:

$$\Theta^m = \text{softmax}\left(\frac{1}{\sqrt{d_k}}\mathcal{G}_Q^m \odot \mathcal{G}_K^m\right) \tag{4}$$

where $\odot$ denotes the element-wise product, $\sqrt{d_k}$ is a scaling factor.

More specifically, the modality $m$ core tensor $\mathcal{G}_K$ and $\mathcal{G}_Q$ are expressed as follows:

$$\begin{cases} \mathcal{G}_Q^m = reshape((\mathbf{Z}^m W_{Q_m}^{(1)}) \otimes_1 (\mathbf{Z}^m W_{Q_m}^{(2)})) \\ \mathcal{G}_K^m = reshape((\mathbf{Z}^m W_{K_m}^{(1)}) \otimes_1 (\mathbf{Z}^m W_{K_m}^{(2)})) \end{cases} \tag{5}$$

where $m \in \{t, a, v\}$, $W_{Q_m}^{(1)} \in \mathbb{R}^{d_m \times r_s}$, $W_{Q_m}^{(2)} \in \mathbb{R}^{d_m \times r_w}$, $W_{K_m}^{(1)} \in \mathbb{R}^{d_m \times r_s}$, $W_{K_m}^{(2)} \in \mathbb{R}^{d_m \times r_w}$ are the linear transformation matrix; $\otimes_1$ denotes the mode-1 Khatri-Rao product.

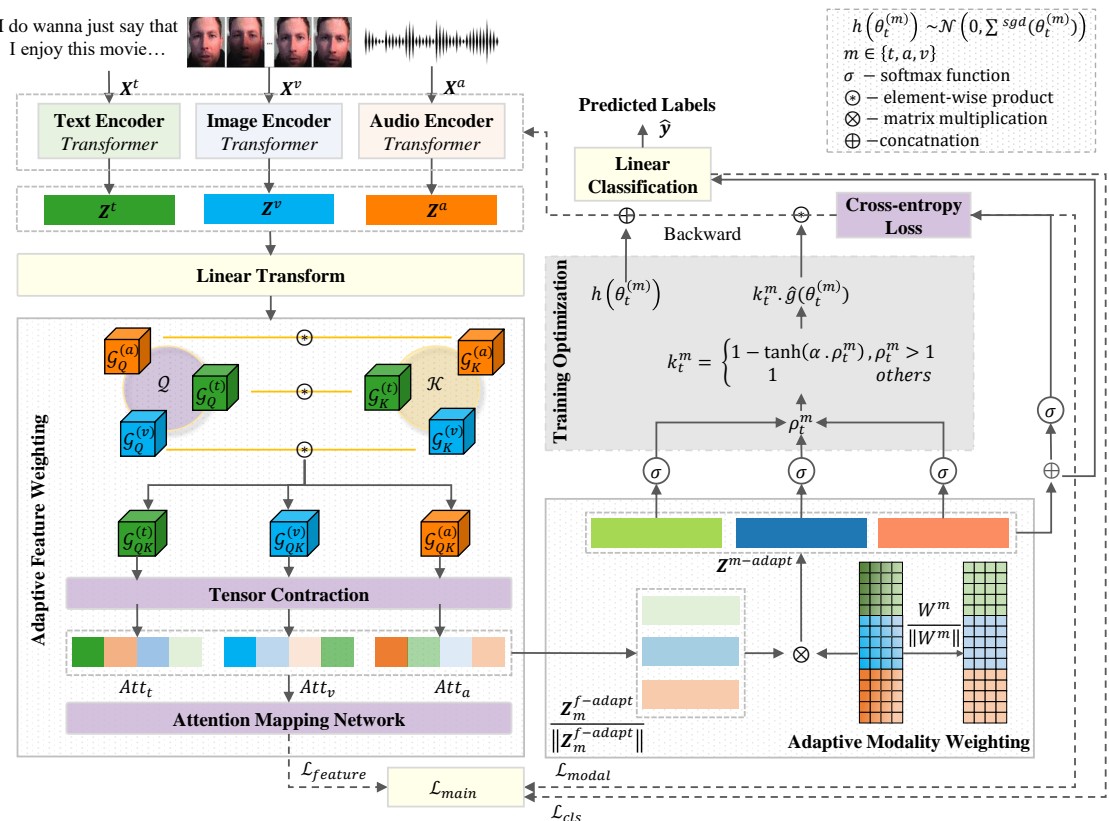

**Figure 2: Illustration of Ada2I framework during the training phase**

*3.2.2 Adaptive Feature Weighting (AFW).* This module addresses the varying impact of each modality on inter-modality and intra-modality interactions using attention mechanism. First, we calculate the attention pooling matrices $\mathbf{A}^{(m)} \in \mathbb{R}^{r_s \times r_w}$ by averaging $\Theta^{(m)}$ across the modality dimension $d_m$, $m \in \{t, a, v\}$. Inspired by MMT [31], the *feature*-aware attention matrix $Att_m \in \mathbb{R}^{N \times d_m}$ for a given modality $m$ is computed as follows:

$$Att_m = \text{Linear}\left(\Theta^m \times_3^1 \mathbf{A}^{(t)} \times_3^1 \mathbf{A}^{(a)} \times_3^1 \mathbf{A}^{(v)}\right) \quad (6)$$

where $\times_3^1$ is the $mode - \binom{1}{3}$ tensor contraction. The *feature*-aware balanced representation $\mathbf{Z}_m^{f-adapt} \in \mathbb{R}^{N \times d_m}$ of the conversation C for a given modality $m$ is computed as:

$$\mathbf{Z}_m^{f-adapt} = Att_m \mathbf{Z}^m + \beta \mathbf{Z}^m \quad (7)$$

where $\beta \in [0, 1]$ is a balancing parameter to regulate the contribution of the original unimodal feature vector $\mathbf{Z}^m$.

## 3.3 Adaptive Modality Weighting (AMW)

Our key focus is to achieve balanced contributions from each modality during the training. Similar to [39], we observe the imbalance problem in multimodal ERC through experiments analyzing the modality-wise weight in norm of each label during training. Apparently, the dominant unimodal encoder, e.g., text, tends to have its

weight in norm increase much faster than the weaker modalities, i.e., audio and visual, leading to divergent unimodal logit scores and distorting the joint fusion representation. Inspired by [34, 43], we propose to incorporate modality-wise L2 normalization to properly weight features, mitigating imbalances arising from differing data distributions and noise levels across modalities. This dynamic adjustment prevents any single modality from dominating the fusion process, thus enhancing overall performance.

Therefore, the modality-level balanced representation $\mathbf{Z}^{m-adapt}$ of the given conversation is calculated as follows:

$$\mathbf{Z}^{m-adapt} = \sum_m^{\{t,a,v\}} \frac{W^m \mathbf{Z}_m^{f-adapt}}{\|W^m\|\|\mathbf{Z}_m^{f-adapt}\|} + b \quad (8)$$

where $W^m \in \mathbb{R}^{d_m \times |\mathcal{E}|}$ symbolizes the output matrix of the model pertaining to modality $m$, and $\mathcal{E}$ is the set of emotion classes.

For **emotion recognition**, we feed $\mathbf{Z}^{m-adapt}$, into the mulilayer preceptron (MLP) with ReLU activation function to compute the output $\hat{y}_i \in \mathbb{R}^{N \times |\mathcal{E}|}$.

$$\hat{y}_i = \text{MLP}(\mathbf{Z}^{m-adapt}) \quad (9)$$

The output $\hat{y}_i$ is utilized to predict emotion labels.

## 3.4 Learning

First, we investigate the standard cross-entropy loss for this downstream task, i.e., mutilmodal ERC as:

$$\mathcal{L}_{cls} = -\frac{1}{B} \sum_{i}^{B} y_i log \hat{y}_i \tag{10}$$

where $B$ is the batch size.

Second, in order to align between the original unimodal representation of modality $m$ and its respective *feature*-aware attention weights as Eq (6), we employ Attention Mapping Network as follows:

$$\hat{Att}_m = \Phi_m(\mathbf{Z}_m, \psi^{(m)}), m \in \{t, a, v\} \tag{11}$$

where $\Phi_m(\cdot)$ is a feed-forward neural network with the parameter $\psi^{(m)}$, $\hat{Att}_m \in \mathbb{R}^{N \times d_m}$ is the *feature*-aware self-attention weights of the modality $m$. To enhance feature-level balance across all modalities, we introduce a L1-norm loss $\mathcal{L}_{feature}$ as:

$$\mathcal{L}_{feature} = \frac{1}{B} \sum_{i}^{B} \left( \sum_{m}^{\{t,a,v\}} |Att_m^i - \hat{Att}_m^i| \right) \tag{12}$$

Additionally, we also consider the modality-level balance loss $\mathcal{L}_{modal}$, which is computed as:

$$\mathcal{L}_{modal} = -\frac{1}{B} \sum_{i}^{B} \log \frac{e^{\mathbf{Z}_i^{m-adapt}}}{\sum_{j=1}^{|\mathcal{E}|} e^{\mathbf{Z}_j^{m-adapt}}} \tag{13}$$

where $\mathbf{Z}_j^{m-adapt}$ represents the output of the $j$-th class for the $i$-th sample. Finally, we combine the all loss functions into a joint objective function, which is used to optimize all trainable parameters in an end-to-end manner.

$$\mathcal{L}_{main} = \mathcal{L}_{modal} + \mathcal{L}_{feature} + \mathcal{L}_{cls} \tag{14}$$

**Training Optimization Strategy**

Recent studies have brought attention to the challenge of handling imbalanced optimization in joint learning models, particularly when dealing with multiple modalities. Peng et al. [25] introduce the OGM-GE method to address optimization imbalances encountered during the simultaneous training of dual-modal systems, i.e., visual and audio. However, directly applying the OGM-GE method to our framework is not practical as it only deals with two modalities. In contrast, our framework caters to more than two modalities across different domains, specifically tailored for the multimodal ERC task. Therefore, leanrable parameter of encoder layer is optimized during training process as the following strategy:

$$\theta_{t+1}^{(m)} = \theta_t^{(m)} - \eta . \hat{g}(\theta_t^{(m)}) \tag{15}$$

where $\hat{g}(\theta^m t) = \frac{1}{o} \sum x \in B_t \nabla_{\theta_t^m} \ell(x, \theta_t^{(i)})$ represents an unbiased estimation of the full gradient $\nabla \theta_t^m \ell(x, \theta_t^{(i)})$ using a random minibatch $B_t$ chosen at the $t$-th step with size $o$. The term $\nabla \theta_t^m \ell(x, \theta_t^{(i)})$ denotes the gradient with respect to $B_t$.

We adjust the balance of modalities through gradient parameter adjustments. For each output at step $t$, we compute the discrepancy ratio for each modality using the softmax of the cosine similarity

---

**Algorithm 1** Ada2I Training Procedure

**Input:** The training set $\mathcal{D} = \{(x_i^t, x_i^a, x_i^v), y_i\}_{i=1}^{N}, m \in \{t, a, v\}$
**Output:** Prediction emotion label $\hat{y}$
**for** each training epoch **do**
  **for** minibatch $\mathcal{B} = \{(x_i^t, x_i^a, x_i^v), y_i\}_{i=1}^{N}\}$ sampled from $\mathcal{D}$ **do**
    #*Refer to Subsection 3.1*
    Encode unimodal feature $\mathbf{X}^m$ to $\mathbf{Z}^m$ as Eq (2)
    #*Refer to Subsection 3.2*
    Multimodal feature representation as Eq (3)
    Calculate coefficient matrix $\Theta^m$ as Eq (4)
    Calculate modality-aware attention $Att_m$ as Eq (6)
    Compute fused feature $\mathbf{Z}_m^{f-adapt}$ with $\beta$ using Eq (7)
    #*Refer to Subsection 3.3*
    Compute logit output $\mathbf{Z}^{m-adapt}$ with modality-wise L2 normalization as Eq (8)
    Produce prediction of multimodal data $\hat{y}_i$ as Eq (9)
    #*Refer to Subsection 3.4*
    Use cross-entropy loss to calculate $\mathcal{L}_{cls}$ as Eq (10)
    Use $L_1$ to calculate $\mathcal{L}_{feature}$ as Eq (12)
    Use cross-entropy to calculate $\mathcal{L}_{modal}$ as Eq (13)
    Add $\mathcal{L}_{feature}$, $\mathcal{L}_{modal}$ and $\mathcal{L}_{cls}$ to compute $\mathcal{L}_{main}$ as Eq (14)
    Compute discrepancy ratio $\rho_t^m = \frac{s_t^m}{\min_{m \in \{t,a,v\}}(s_t^j)}$
    Compute modulation coefficient $k_t^m$
    Update using $\theta_{t+1}^{(i)} = \theta_t^{(i)} - \eta \cdot \hat{g}(\theta_t^{(i)}) \cdot k_t^i + \eta \cdot h(\theta_t^{(i)})$
  **end for**
**end for**

---

between the output weights and the corresponding feature vectors:

$$s_t^m = \sum_{j=1}^{L} \sum_{k=1}^{\mathcal{E}} \mathbb{I}_{k=y_j} \text{softmax}(cos\langle W_k^m, \mathbf{Z}_k^m \rangle + \frac{b_k}{M}) jk \tag{16}$$

where $\mathbb{I}_{k=y_j}$ equals 1 if $k = y_j$ and 0 otherwise, and softmax(.) estimates the unimodal performance of the multimodal model, $M$ denotes the count of modalities. Specifically, for the Multimodal ERC task under consideration, we delineate three modalities: text $(t)$, audio $(a)$, and visual $(v)$. The discrepancy ratio is calculated as:

$$\rho_t^m = \frac{s_t^m}{\min_{m \in \{t,a,v\}}(s_t^j)} \tag{17}$$

The learnable parameters are updated according to:

$$\theta_{t+1}^{(m)} = \theta_t^{(m)} - \eta . \hat{g}(\theta_t^{(m)}) . k_t^m \tag{18}$$

where the modulation coefficient $k_t^m$ is determined by $1 - \tanh(\alpha \cdot \rho_t^m)$ if $\rho_t^m > 1$, and 1 otherwise. Here, $\alpha$ is a hyperparameter controlling the degree of modulation. Additionally, to enhance the adaptability of the modulation process, Gaussian noise $h(\theta_t^{(i)})$ sampled from a distribution $\mathcal{N}(0, \sum^{sgd}(\theta_t^{(i)}))$ is introduced after parameter updates:

$$\theta_{t+1}^{(i)} = \theta_t^{(i)} - \eta \cdot \hat{g}(\theta_t^{(i)}) \cdot k_t^i + \eta \cdot h(\theta_t^{(i)}) \tag{19}$$

The training process of Ada2I is illustrated in Algorithm 1.

# 4 EXPERIMENTIAL SETUP

## 4.1 Datasets

*Datasets:* We consider three benchmark datasets for multimodal ERC namely: IEMOCAP [3], MELD [27], and CMU-MOSEI [1]. The dataset statistics are illustrated in Table 1.

*IEMOCAP.* This datasset offers 12 hours of video recordings capturing dyadic conversations involving 10 speakers. Each video contains a single dialogue, segmented into utterances, resulting in a total of 7,433 utterances across 151 dialogues. Notably, each utterance is annotated with one of six emotion labels: happy, sad, neutral, angry, excited, or frustrated.

*MELD.* This dataset is based on the TV series Friends, includes 13,709 video clips featuring multi-party conversations, each labeled with one of Ekman's six universal emotions: joy, sadness, fear, anger, surprise, and disgust.

*CMU-MOSEI:.* This dataset is a prominent resource for sentiment and emotion analysis, comprises 3,228 YouTube videos divided into 23,453 segments, featuring contributions from 1,000 speakers covering 250 topics. It includes six emotion categories: happy, sad, angry, scared, disgusted, and surprised, with sentiment intensity ranging from -3 to 3.

**Table 1: Data Statistics**

| Datasets | Dialogues | | | Utterances | | |
|---|---|---|---|---|---|---|
| | train | valid | test | train | valid | test |
| IEMOCAP | 120 | | 31 | 5,810 | | 1,623 |
| MELD | 1,039 | 114 | 280 | 9,989 | 1,109 | 2,610 |
| CMU-MOSEI | 2,248 | 300 | 676 | 16,326 | 1,871 | 4,659 |

## 4.2 Baselines and Evaluation Metrics

*Baselines:* Ada2I is compared against several state-of-the-art (SOTA) baseline approaches for evaluating performance in multimodal ERC, particularly addressing modality imbalance problems. For the IEMOCAP and MELD datasets, we consider baseline models such as DialogueRNN [23], DialogueGCN [8], MMGCN [37], BiD-DIN [40], and MM-DFN [11]. We report the best results obtained from [36], which enhanced these models to address modality imbalance. Additionally, we consider other SOTA models for multimodal ERC that do not explicitly address modality imbalance, including COGMEN [16], CORECT [24], GraphMFT [18], DF-ERC [17], and AdaIGN [32]. For the CMU-MOSEI dataset, we evaluated various baseline models for sentiment classification tasks, which include both 2-class sentiment, featuring only positive and negative sentiment, and 7-class sentiment, ranging from highly negative (-3) to highly positive (+3). These baseline models include Multilouge-Net [30], TBJE [4], COGMEN [16], CORECT [24], OGM-GE [25], and $I^2$MCL [44]. Notably, OGM-GE and $I^2$MCL specifically address the issue of imbalanced modalities in multimodal ERC, whereas the other baseline models do not.

*Evaluation Metrics:* Similar to prior studies [23, 36, 37], we evaluate the effectiveness of emotion recognition using Accuracy (Acc) and Weighted F1 Score (WF1) as our primary evaluation metrics.

## 4.3 Experimental Settings

We derive multimodal features for each utterance from acoustic, lexical, and visual modalities using a combination of models and pre-trained models, as outlined in Table 2. We employ PyTorch[1] for training our architecture and Comet[2] for logging all experiments, leveraging its Bayesian optimizer for hyperparameter tuning. Additional parameters can be found in Table 2.

**Table 2: Hyper-parameter settings**

| Parameter/Module | IEMOCAP | MELD | CMU-MOSEI |
|---|---|---|---|
| Text Feature Extraction | sBERT[3] | | |
| Audio Feature Extraction | Wave2vec-Large [28], OpenSmile [5] | | |
| Visual Feature Extraction | MTCNN [41], MA-Net[4], DenseNet [12] | | |
| Text embedding dim. $d_t$ | 768 | 768 | 768 |
| Audio embedding dim. $d_a$ | 512 | 300 | 512 |
| Visual embedding dim. $d_v$ | 1024 | 342 | 1024 |
| hidden dim | 300 | 200 | 500 |
| tensor rank | 11 | 6 | 10 |
| $\alpha$ | 0.037 | 0.4 | 0.4 |
| $\beta$ | 0.01 | 0.55 | 0.2 |
| learning rate | 1.7e-4 | 1.2e-4 | 1.9e-4 |
| batch size | 10 | 10 | 32 |
| epoch | 50 | 50 | 30 |

# 5 RESULTS AND DISCUSSION

## 5.1 Performance Comparison against Baselines

*IEMOCAP and MELD dataset:* As depicted in Table 3, our model Ada2I performs better than the previous SOTA baselines in the context of balanced modality consideration on all modality combinations on both datasets. Indeed, in the AV modality pair on the MELD dataset, traditionally deemed the weakest, we observe a substantial performance boost in Multimodal ERC. Specifically, there is a noteworthy enhancement of 10.77% on WF1 and 6.98% on Accuracy compared to the previous SOTA model. This progress effectively reduces the performance discrepancy compared to modality pairs where text plays a dominant role.

We also compare Ada2I with SOTA baseline models for multimodal ERC, particularly those focusing solely on multimodal fusion and architectural design without addressing modality imbalance. Figure 3b demonstrates that our proposed Ada2I significantly reduces the performance gap in WF1 between learning from all three modalities simultaneously (T+A+V) and pair-wise modality combinations on the MELD dataset. Most notably, with the weaker

---

[1]https://pytorch.org/
[2]https://comet.ml
[3]https://www.sbert.net/
[4]https://github.com/zengqunzhao/MA-Net

**Table 3: Comparison results in the multimodal setting of Ada2I with the modal-balanced baseline model enhanced by FAGM [36]. The best performance is marked in bold, and the second best performance is indicated by underlining.**

| Methods | IEMOCAP | | | | | | | | MELD | | | | | | | |
|---|---|---|---|---|---|---|---|---|---|---|---|---|---|---|---|---|
| | TAV | | TA | | TV | | AV | | TAV | | TA | | TV | | AV | |
| | W-F1 | Acc | W-F1 | Acc | W-F1 | Acc | W-F1 | Acc | W-F1 | Acc | W-F1 | Acc | W-F1 | Acc | W-F1 | Acc |
| DialogueRNN† | 61.31 | 61.61 | 61.90 | 61.98 | 60.19 | 59.95 | 48.31 | 50.71 | 56.42 | 58.05 | 56.46 | 58.01 | 55.67 | 57.39 | 40.46 | 45.39 |
| DialogueGCN† | 62.76 | 63.22 | 64.36 | 64.39 | 61.25 | 62.23 | 49.20 | 49.85 | 54.61 | 58.96 | 54.80 | 57.28 | 55.26 | 57.10 | 10.02 | 44.44 |
| BiDDIN† | 58.81 | 58.84 | 58.88 | 58.16 | 59.04 | 58.96 | 46.36 | 46.77 | 57.47 | 59.18 | 56.56 | 58.05 | 56.93 | 58.10 | 44.39 | 48.62 |
| MM-DFN† | 64.92 | 64.57 | 63.91 | 64.20 | 61.02 | 60.60 | 54.48 | 55.03 | 55.75 | 60.8 | 57.10 | 60.00 | 57.73 | 60.65 | 42.05 | 48.66 |
| MMGCN† | 64.53 | 64.51 | 63.25 | 63.40 | 61.02 | 61.06 | 54.14 | 54.90 | 58.48 | 61.15 | 57.59 | 60.69 | 57.14 | 59.46 | 43.49 | 48.43 |
| **Ada2I** | **68.97** | **68.76** | **66.91** | **67.28** | **65.48** | **65.43** | **55.16** | **55.64** | **60.38** | **63.03** | **60.08** | **62.64** | **58.62** | **61.95** | **55.16** | **55.64** |
| Δ | ↑4.05 | ↑4.19 | ↑2.55 | ↑2.89 | ↑4.23 | ↑3.20 | ↑0.68 | ↑0.61 | ↑1.90 | ↑1.88 | ↑2.49 | ↑1.95 | ↑0.89 | ↑1.30 | ↑10.77 | ↑6.98 |

modality pair (audio+visual) consistently lagging behind in performance compared to the full modality combination (i.e., with AdaIGN, this gap is 23.12%), Ada2I boosts the model and shortens the gap to only 5.22%. Similarly, with the text+audio (T+A) and text+visual (T+V) pairs, this gap is also substantially reduced, indicating that the model has learned in a more balanced manner, leveraging additional useful information from non-dominant modalities. The significant improvement is similarly observed on the IEMOCAP dataset in Figure 3a.

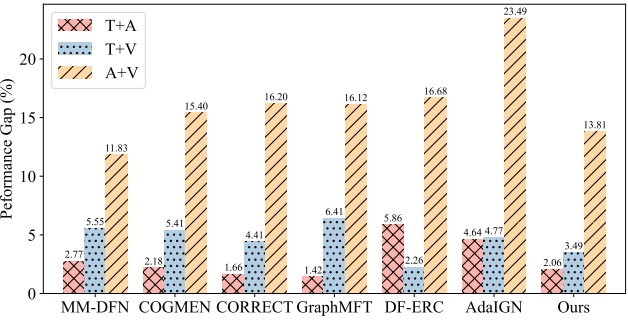

**(a) on IEMOCAP dataset**

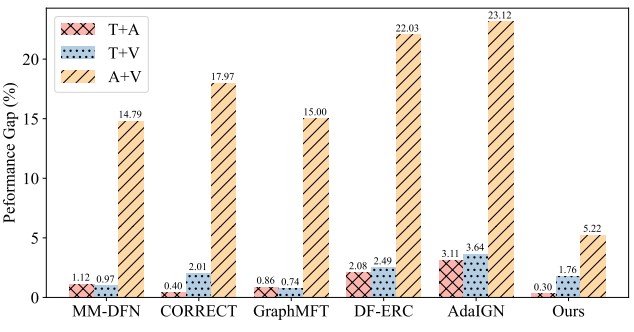

**(b) on MELD dataset**

**Figure 3: Performance gap visualizations between the multimodal setting (T+A+V) and pair-wise modality combinations are evaluated using the WF1 metric across the IEMOCAP and MELD datasets.**

**Table 4: Results on the CMU-MOSEI dataset. The best performance is highlighted in bold. For cells displaying "-", the results were not provided in the paper. † denotes results obtained from running the provided code in original paper.**

| Methods | 2-class | | | | 7-class | | | |
|---|---|---|---|---|---|---|---|---|
| | TAV | TA | TV | AV | TAV | TA | TV | AV |
| Multilouge-Net [30] | 82.10 | 80.18 | 80.06 | **75.16** | 44.83 | - | - | - |
| TBJE [4] | 81.50 | 82.40 | - | - | 44.40 | 45.50 | - | - |
| COGMEN† [16] | 82.95 | 85.00 | 82.99 | 65.95 | 43.90 | 44.31 | 42.68 | 24.27 |
| CORECT† [24] | 83.98 | 84.28 | 82.83 | 68.89 | 46.31 | 44.89 | 43.76 | 24.55 |
| I²MCL [44] | 81.05 | - | - | - | 45.43 | 43.68 | 44.44 | 31.53 |
| OGM-GE† [25] | 84.58 | 84.03 | 83.67 | 71.53 | | | | |
| **Ada2I** | **85.25** | **85.08** | **85.21** | 74.93 | **47.71** | **47.35** | **47.37** | **34.64** |
| Δ | ↑0.67 | ↑0.08 | ↑1.54 | ↓0.23 | ↑2.28 | ↑1.85 | ↑2.93 | ↑3.11 |

*CMU-MOSEI dataset:* Table 4 shows that Ada2I outperforms all baseline models. Specifically, when compared to OGM-GE and I²MCL, two models proposed for addressing modality imbalance during training, Ada2I demonstrates superior performance across all modality combinations. When compared to other baseline models that do not consider modality balancing, Ada2I also demonstrates significant balancing capabilities, reducing the performance gap between modality pairs. For instance, in the CORECT model, the gap between T+A+V and A+V is 15.09% for 2-class sentiment, and this figure increases to 21.76% for 7-class sentiment. However, with Ada2I, these gaps are significantly reduced to 10.32% and 13.07%, respectively, underscoring the effectiveness of Ada2I in addressing modality imbalances.

## 5.2 Ablation Study

*5.2.1 Balancing Interpretation.* We conduct ablation studies with the two main modules of the model, AMW and AFW, to assess their impact on the Ada2I model. Additionally, through the Discrepancy Ratio, we interpret the model's balancing by observing its changes. A smaller Discrepancy Ratio indicates a more balanced optimization process. Figure 4 shows that the discrepancy ratios $\rho^t$, $\rho^v$, and $\rho^a$ significantly decrease when both AMW and AFW are combined within Ada2I, with all ratios approaching approximately 1 on the IEMOCAP dataset. In contrast, when one of the modules is ablated,

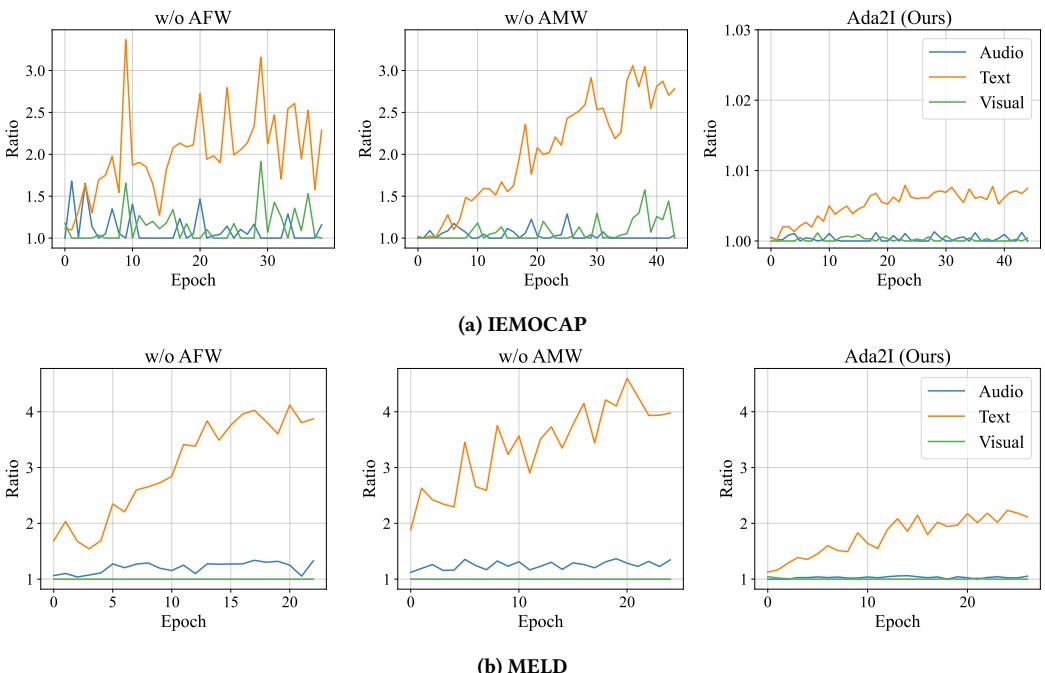

(a) IEMOCAP

(b) MELD

Figure 4: The change of the discrepancy ratio $\rho^t, \rho^a, \rho^v$ on the IEMOCAP and MELD datasets during training, along with various ablation tests including without AMW and without AFW, are compared to the Ada2I model.

the ratios for audio ($\rho^a$) and visual ($\rho^v$) are approximately 1.5, while for text, it increases to around 3. Similarly, on the MELD dataset, our proposed model Ada2I has reduced this discrepancy ratio of text from over 4 (w/o AFW) to approximately half, reaching around 2, while for audio and visual, it brings them close to the 1 mark. In summary, the combined design of both modules AMW and AFW enhances balanced learning across modalities during training, highlighting the significance and inseparability of feature-level and modality-level balancing.

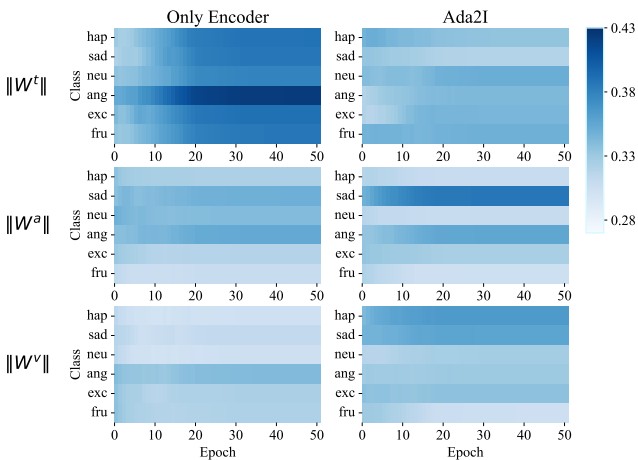

Figure 5: The observation of modality-wise weights for each label normalized for the IEMOCAP dataset

*5.2.2 Effect of Weight Normalization.* As mentioned earlier, the unimodal weights also directly influence the encoder updating process. The imbalanced weight components induce gradients and subsequently lead to the inconsistent convergence of unimodalities. Here, we provide a clearer visualization of these unimodal weights before imbalance processing (Only Encoder) and in the Ada2I model in Figure 5 for the IEMOCAP dataset. It is evident that with Only Encoder, the text encoder (dominant modality) weight in norm grows much faster than audio and visual. After balancing, our model exhibits a more balanced optimization process.

## 6 CONCLUSION

In this work, we present **Ada2I**, a framework designed to address modality imbalances and optimize learning in multimodal ERC. We identify and analyze existing issues in current ERC models that overlook the imbalance problem. From there, we propose a solution comprising integral modules: Adaptive Feature Weighting (AFW) and Adaptive Modality Weighting (AMW). The former enhances intra-modal representations for feature-level balancing, while the latter optimizes inter-modal learning weights with the balancing at modality level. Furthermore, we introduce a refined disparity ratio to optimize training, offering a straightforward yet effective measure to evaluate the model's overall discrepancy when handling multiple modalities simultaneously. Extensive experiments on the IEMOCAP, MELD, and CMU-MOSEI datasets validate its effectiveness, showcasing SOTA performance. In the future, we anticipate enhancing the efficiency of the framework and maximizing the utilization of emotional cues.

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
