# OpenReview forum: "Ada2I: Enhancing Modality Balance for Multimodal Conversational Emotion Recognition"
_acmmm.org/ACMMM/2024/Conference — MM2024 Poster_

### Official Review · Reviewer_RJYY · 2024-05-10

**Rating:** 3
**Confidence:** 4

**Summary:**

This paper aims to address the issue of modality imbalance in multimodal learning, commonly known as modality collapse. To tackle this problem, the authors employ Adaptive Feature Weighting (AFW) to achieve feature-level balance and Adaptive Modality Weighting (AMW) for modality-level balancing. In addition, the authors optimize the training optimization strategy.  Experiments were conducted on a wide range of datasets.

**Strengths:**

1. The paper introduces a new framework called Ada2I, focusing on addressing modality imbalance in MERC. The design of AFW and AMW allows the model to balance at both the feature and modality levels.
2. Extensive experiments on three different benchmark datasets in the paper validate the effectiveness of the model.
3. The paper provides a novel training optimization strategy that helps to better understand and control the modality imbalance problem during the model training process.

**Limitations:**

1. The proposed improvements, while addressing the problem from two perspectives, seem to lack clear innovation as they mainly integrate existing methods without showcasing distinctiveness. As a result, the contribution appears to be limited.

2. The main motivation is the modality imbalance problem in multimodal joint training. What would be the outcome if each modality underwent separate training, followed by integration of their results? Is the lack of significant improvement in multimodal performance due to individual modalities' inadequate performance or incomplete learning stemming from joint training? Many MLLMs involve mapping other modalities into the textual space for problem-solving, such as LLava and mini-GPT4. In such a context, what is the significance of the modality imbalance issue? What are its practical implications and application scenarios?

3. While the paper compares the performance of Ada2I with state-of-the-art models on the IEMOCA and MELD datasets, a direct performance comparison with SOTA is not given in Table 3. Although Figure 3 provides visualisation of the performance gap, these results alone do not convincingly demonstrate that solving the modal imbalance problem significantly improves performance. This shortcoming is also evident in Table 4, where the performance improvement of Ada2I on the MOSEI dataset is limited even after accounting for the modal imbalance problem.


4. The paper lacks certain experimental details, which may hinder result reproducibility. For example, the division of the IEMOCAP dataset is not explicitly outlined. Was it divided based on different sessions?

5. The explanations regarding how features and modality weights impact emotion recognition results in the ablation study are insufficient.

6. The parameter $\beta$ in Equation 7 seems significant, however, its influence on the final performance remains unclear. The ablation of some other hyperparameters is also lacking.

7. There are some issues in the writing and presentation. The motivation outlined in the introduction is not sufficiently clear, and discrepancies in Figure 1 undermine the credibility of the experiments. It's advisable for the author to verify the accuracy of the annotations. Furthermore, Section 2.1 of the related work enumerates numerous Graph-based MERC methods, which seem tangential to the approach proposed in this paper. How does the MERC scenario differ from other tasks? Is it inherently unique? Why does it necessitate specialized design, and what sets apart the method proposed in this paper from previous works? These aspects are not adequately addressed in the related work section. The methodology section suffers from symbol confusion, exemplified by the repeated use of 'M' in line 277 without clear distinction in meaning.

**Suitability:**

3

---

### Official Review · Reviewer_Q5rm · 2024-05-24

**Rating:** 3
**Confidence:** 3

**Summary:**

This work presents a new approach for multimodal conversational emotion recognition (MER). It involves two attempts from both feature view and modality view. Meanwhile, an enhancement for training is also proposed.

**Strengths:**

1. The work aims at improving MER task from an overall view, and constructs learnable weights from both feature and modality prospects.
2. The end-to-end framework of Ada2I is complete.

**Limitations:**

1. The illustration of $\mathcal{G}$ is included in an cited article [42] implicitly, which is a basic component for the whole process. And it seems that $\mathcal{G}_{Q, K}^{(m)}$ is obtained from a linear transform via Fig.2. It's suggested to add a more detailed illustration about that. Besides, I'm confused about the relationship between $(d_a, d_v)$ in Line 320 and $(r_s, r_w)$ in Line 330. Does $(s, w) \in \{1, 2, 3\}$ means different modalities from $(t, a, v)$?
2. Only impacts of AFW and AMW modules are considered in ablation studies, which lack the analyze of training optimization strategy.

**Suitability:**

3

---

### Official Review · Reviewer_Th2d · 2024-05-24

**Rating:** 5
**Confidence:** 4

**Summary:**

This paper introduces a novel framework, Ada2I, designed to address the challenges of modality imbalances in multimodal emotion recognition in conversations (ERC). The authors propose a two-module system: Adaptive Feature Weighting (AFW) for feature-level balancing and Adaptive Modality Weighting (AMW) for modality-level balancing. They also introduce a refined disparity ratio as a training optimization strategy. The Ada2I framework is evaluated on three benchmark datasets: IEMOCAP, MELD, and CMU-MOSEI, demonstrating state-of-the-art performance and effective handling of modality imbalances.

**Strengths:**

(1) The paper tackles a significant problem in multimodal ERC, which is the imbalance in contributions from different modalities and proposes a comprehensive solution with the Ada2I framework.
(2) The introduction of the Adaptive Feature Weighting and Adaptive Modality Weighting modules provides a systematic approach to balancing feature-level and modality-level representations, respectively.
(3) The refined disparity ratio as a training optimization strategy is a novel contribution that offers a clear measure to evaluate the model's learning process across multiple modalities.
(4) The paper provides extensive experimental results, demonstrating the effectiveness of Ada2I against several state-of-the-art baselines on three well-known benchmark datasets.
(5) The ablation studies and visualizations of the discrepancy ratio and modality-wise weights provide insightful analysis of the model's performance and the impact of the proposed modules.

**Limitations:**

(1) The paper could benefit from a more detailed explanation of the tensor-ring decomposition method and its integration into the model, as this is a complex and less commonly used technique.
(2) While the paper mentions the use of pretrained models for feature extraction, it lacks specifics on how these models were chosen and whether their pretrained weights were fine-tuned for the task.
(3) The paper could provide more insight into the selection of hyperparameters and whether there were any sensitivity analyses conducted to understand their impact on model performance.
(4) Although the paper presents improvements over baseline models, it is not clear how Ada2I compares to models that do not focus on modality imbalance but have different architectural designs or fusion techniques.

**Suitability:**

3

---

### Official Review · Reviewer_C26X · 2024-05-25

**Rating:** 2
**Confidence:** 3

**Summary:**

The paper introduces Ada2I, a novel framework specifically designed to tackle the challenge of modality imbalance in MERC. The authors address this challenge by implementing Adaptive Feature Weighting (AFW) for feature-level balance and Adaptive Modality Weighting (AMW) for modality-level balancing. Additionally, they introduce a refined disparity ratio to optimize the training process. The framework's effectiveness is demonstrated through extensive experiments on three benchmark datasets: IEMOCAP, MELD, and CMU-MOSEI.

**Strengths:**

- The paper designs two integral modules, AFW and AMW, which operate in tandem to ensure balanced learning across modalities.
- The paper validates the effectiveness of Ada2I against several baselines, showcasing improved performance on benchmark
 datasets.
- The paper is well-written, with a clear presentation of methodology.

**Limitations:**

- Lack of novelty. The proposed improvements mainly integrate existing methods without demonstrating distinctiveness and their contribution is limited.
- The motivation is rather lacking for the modal imbalance problem. Some of the work maps other modalities to text space to solve the problem. Then the significance of the modal imbalance solution should be more fully explained in the text. Also, do the methods chosen in the experiments take into account the modal imbalance problem, and are there methods that do not take the above into account and still perform better?
- The ablation experiments need to be supplemented. Some of the parameters in the methods lack relevant discussion. Some of the hyperparameters may have a more significant effect on the final results.

**Suitability:**

3

---

### Meta-Review · Area_Chair_hC4M · 2024-07-03

**Recommendation:** Accept (Poster)
**Confidence:** 4

**Metareview:**

Test-Time Adaptation for Multimodal Sentiment Analysis